# A Survey of Changes in the Psychological State of Individuals with Social Withdrawal (*hikikomori*) in the Context of the COVID Pandemic

Takafumi Ogawa [1], Yuki Shiratori [2,*], Haruhiko Midorikawa [3], Miyuki Aiba [4], Daichi Sugawara [5], Naoaki Kawakami [5], Tetsuaki Arai [2] and Hirokazu Tachikawa [6]

1   Ibaraki Prefectural Medical Center of Psychiatry, Asahi654, Kasama 309-1717, Japan
2   Department of Psychiatry, Faculty of Medicine, University of Tsukuba, 1-1-1 Tennoudai, Tsukuba 305-8575, Japan; 4632tetsu@md.tsukuba.ac.jp
3   Department of Psychiatry, University of Tsukuba Hospital, 2-1-1 Amakubo, Tsukuba 305-8575, Japan
4   Faculty of Human Sciences, Toyo Gakuen University, 1-26-3 Hongo, Bunkyo, Tokyo 113-0033, Japan
5   Faculty of Human Sciences, University of Tsukuba, 1-1-1 Tennoudai, Tsukuba 305-8575, Japan; sugawara@human.tsukuba.ac.jp (D.S.); kawakami.naoaki.fn@u.tsukuba.ac.jp (N.K.)
6   Department of Disaster and Community Psychiatry, Division of Clinical Medicine, Faculty of Medicine, University of Tsukuba, 1-1-1 Tennoudai, Tsukuba 305-8575, Japan; tachikawa@md.tsukuba.ac.jp
*   Correspondence: yuki.shiratori@md.tsukuba.ac.jp

**Abstract:** Background: The coronavirus disease (COVID) 2019 pandemic has been reported to have resulted in psychological disturbances. The Japanese term "hikikomori" refers to a state of preferring to stay at home. The COVID pandemic provided an opportunity to extend our current understanding of hikikomori by examining the psychological states of individuals who were in this state under lockdown, during which, paradoxically, their condition may have been adaptive. Methods: We administered a questionnaire to examine psychological changes among 600 people with hikikomori traits in Japan. The Hospital Anxiety and Depression Scale (HADS), Subjective Stress Scale, and Stigma Questions for hikikomori were administered retrospectively at three time points. We also collected descriptive data regarding the participants' coping strategies. Results: The participants' sense of stigma regarding hikikomori was improved during the pandemic, whereas depression and anxiety worsened. The participants with 'definite' hikikomori (they met the diagnostic criteria) reported more severe depression and anxiety than those with 'possible' hikikomori. Their coping strategies were adaptive to the pandemic situation. Conclusion: Although the sense of stigma against hikikomori was improved and adaptive strategies were employed, the participants with hikikomori experienced a worsening of depression and anxiety during the COVID lockdowns. The improvement of stigma and the participants' indoor adaptive coping strategies could not ameliorate the mental state of hikikomori. Therapeutic interventions should be considered in the future for definite hikikomori meeting the criteria.

**Keywords:** hikikomori; social withdrawal; lockdown; depression; anxiety; stigma; coping method

## 1. Introduction

Coronavirus disease 2019 (COVID-19) has spread worldwide, and numerous studies have examined the mental health impacts of the COVID-19 pandemic, describing the impacts of fear and anxiety about COVID-19 on mental health, as well as the mental health effects of global lockdown policies [1,2]. A systematic review of international studies reported that individuals and populations worldwide experienced a high burden of mental health problems, including depression, anxiety disorders, sleep disorders, post-traumatic stress symptoms, and suicidal behavior [3,4]. It was also suggested that excessive health information about COVID-19 might increase the perceived impact of the pandemic, resulting in worsening mental states [5]. The strict and prompt COVID-19 measures

implemented by some countries' governments have proven to be important in alleviating not only physical but also mental health problems [6].

In Japan, extensive restrictions on activities requested by the government, e.g., social distancing and refraining from going out as countermeasures to infection, were reported to have led to exacerbated employment difficulties and an increased number of suicides in 2020 [7]. This finding suggests that the national confinement in Japan, which induced a state of preferring to stay at home (known as "hikikomori"), had a significant impact on mental health in the Japanese population. Indeed, a study confirmed that many mental health problems occurred during the pandemic and indicated that singleness or separation from other people were risk factors [8].

Hikikomori, which was originally reported in Japan, has attracted international attention in recent years, not only from mental health professionals and researchers, but also from the general public [9]. Hikikomori is not simply a case of social withdrawal but rather a pathological condition that can lead to severe psychological impairment [10]. Hikikomori refers to a behavior pattern in which a person stays at home and withdraws from social activities and interactions, and this behavior pattern has received public attention in Japan since the late 1990s. Although hikikomori was once regarded as a pathological condition of adolescence, it has been observed in all generations and in various countries, including Hong Kong, Spain, India, and the United States [11–14]. A survey conducted in 2015 and 2018 by the Cabinet Office in Japan estimated that the total population of individuals with hikikomori is expected to reach 1,000,000 [15,16]. Providing support to people with hikikomori is expected to become more difficult with the aging of family members who care for them [17].

There have been few investigations of psychological factors related to individuals who have hikikomori. A relationship between hikikomori and narcissism has been suggested; it was reported that people who have hikikomori tend to have traits characterized by "hypervigilant-type" narcissistic tendencies, in which they are hypersensitive to others' evaluations and avoid public appearances for fear of shame [18]. A sense of stigma regarding hikikomori was also recognized as an important factor in exacerbating the phenomenon [19], with one study finding that hikikomori was associated with stigma-related posts on social networking sites [20]. Indeed, hikikomori was repeatedly described by Japanese media in the early 2000s. Now that it is clear that hikikomori may be associated with hypervigilant-type narcissism and that it was sensationalized by media in the past, it is possible that the stigma regarding "hikikomori" might be related to individuals' behavior and mental health. The results of our present study provide new insights into the psychological factors of hikikomori, and since they have not been verified before, our findings might be a keystone in considering ways to provide support for people with hikikomori.

To the best of our knowledge, no studies have examined psychological changes among individuals with hikikomori since the spread of COVID-19 and the subsequent lockdown policies. It is possible that the stigma regarding hikikomori improved following the COVID-19 pandemic, and, paradoxically, the mental health of those with the condition may have improved. It has also been pointed out that the social distancing induced by the infectious COVID outbreak might have had a positive effect by increasing the understanding of hikikomori in the general population [21,22]. We hypothesized that people with hikikomori experience stigma, and that this stigma was reduced during the COVID lockdown period, resulting in increased self-evaluation and improved mental health. To test this hypothesis, we conducted a web-based survey of individuals who had already exhibited a tendency toward hikikomori. We compared the perceived degree of stigma regarding hikikomori and changes in the participants' severity of depression and anxiety before and after the lockdown period. In order to deepen our understanding of hikikomori and learn about the coping strategies of people with this condition, we also conducted a descriptive analysis asking how the participants with hikikomori survived the difficult situations under the lockdown.

## 2. Participants and Methods

### 2.1. Participants

We administered the questionnaire that was used by the Japanese Cabinet in 2018 to survey quality of life among Japanese citizens [14]. The questionnaire asked, "How often do you currently go out?" with the following possible responses: "1. Going out every day for work or school", "2. Going out 3–4 times per week for work or school", "3. Going out frequently for recreational activities and the like", "4. Going out sometimes for social activities", "5. Usually staying at home but going out for hobbies", "6. Usually staying at home but going out to a nearby convenience store", "7. Come out of own room but do not leave home", and "8. Do not come out of own room". We divided these responses into eight levels and the participants who responded at level 5 or higher were included in this study.

This study was conducted as an internet survey in collaboration with Rakuten Insight, Inc., a web research firm, between 11 and 12 August 2020. Rakuten Insight has a large pool of registered Japanese citizens who regularly participate in surveys conducted in Japan. Rakuten Insight used a selective process for the present study, drawing a sample from their extensive pool of respondents. The sample consisted of 4759 participants in their 20s through 50s, and students and housewives were excluded. From this sample, we selected the 600 individuals who met the level ≥5 criterion for further analysis (*n* = 4759, excluded = 4159, selection rate = 12.6%).

### 2.2. Procedures

We divided the 600 participants into those with 'definite hikikomori' and 'possible hikikomori' according to the evaluation criteria of hikikomori developed by Kamba et al. [23]. We defined hikikomori as fulfilling the following three criteria: (1) Marked social isolation in one's home; (2) Duration of continuous social isolation for ≥6 months; and (3) Significant functional impairment or distress with social isolation. In the criteria, the severity of hikikomori was classified based on the frequency of leaving home, where those who occasionally left their home (2–3 days per week) were judged as having mild hikikomori, those who rarely left their home (≤1 day per weeks) were considered to have moderate hikikomori, and individuals who rarely left their single room were deemed to have severe hikikomori. Elderly individuals and homemakers were also considered to have hikikomori.

In our study, we defined individuals who met all three of the above-described criteria as having 'definite' hikikomori, and those who met only Criterion (1) as having 'possible' hikikomori. We decided that the frequency of going out would not be considered in this study, and we chose to focus on the core pathological group consisting of young and middle-aged individuals. To measure Criterion (1), we used a questionnaire that had been issued by the Japanese Cabinet in 2018 to assess the quality of life among Japanese citizens. Participants who met the criterion of scoring at level ≥5 on the questionnaire were included in this study. Criterion (2) was assessed through a question that asked participants if they had experienced social isolation in their home for >6 months, and Criterion (3) was measured using a question that inquired about the presence of severe stress during the period of social isolation within one's home.

To assess psychological changes before and after the COVID-19 pandemic, we administered the Hospital Anxiety and Depression Scale (HADS) for detecting depression and anxiety. We also administered a scale that measures perceived stigma toward hikikomori, which was developed with reference to a scale measuring stigma toward mentally ill people that was devised by Link [24–26].

The questionnaires were cross-sectionally measured as of 11–12 August 2020, and the participants were asked to answer them retrospectively, reflecting on three specific time points: before the issuance of the lockdown order in Japan on 7 April 2020, during the lockdown order (7 April–25 May 2020), and after the end of the lockdown period (25 May–11 or 12 August 2020). By the end of May 2020, a total of 16,851 infection cases (with

the restriction level classified as yellow) and 891 deaths had been reported in Japan. The stringency index during this period ranged from 45.37 to 40.74 [27].

In Study 1, we analyzed the factor structure of the scale that we developed. In Study 2, we analyzed how stigma towards hikikomori and psychological symptoms (depression, anxiety, subjective stress) changed between before and after the COVID-19 pandemic. In Study 3, we conducted a morphological analysis of the free-text section of the question-naires to determine what coping behaviors were employed during the lockdown period by individuals who already had hikikomori traits. We sought to understand how individuals with hikikomori survived and coped with the difficult situation caused by the COVID-19 pandemic.

*2.3. Measures*

2.3.1. Hospital Anxiety and Depression Scale (HADS)

The HADS was developed by Zigmond and Snaith in 1983 to detect and identify anxiety disorder and depression among individuals admitted to non-psychiatric hospitals [28]. This self-assessment scale is verified to be reliable for detecting mental disorders such as those described above. The HADS is divided into two sections (an Anxiety subscale and a Depression subscale) and has been reported to be useful in screening for anxiety and depression separately. Each section consists of seven items, resulting in a total of 14 questions. The responses are rated on a 4-point scale (even though the responses are varied depending on each question, the responses likely follow: 0, Not at all; 1, From time to time, occasionally; 2, A lot of the time; 3, Most of the time: 0, Definitely as much; 1, Not quite so much; 2, Only a little; 3, Hardly at all: 0, Not at all; 1, A little, but it doesn't worry me; 2, Yes, but not too badly; 3, Very definitely and quite badly: 0, As much as I always could; 1, Not quite so much; 2, Definitely not so much now; 3, Not at all: 0, Only occasionally; 1, From time to time, but not too often; 2, A lot of time; 3, A great deal of the time; 0, Most of the time; 1, Sometimes; 2, Not often; 3, Not at all: 0, Definitely; 1, Usually; 2, Not often; 3, Not at all: 0, Not at all; 1, Sometimes; 2, Very often; 3, Nearly all the time: 0, Not at all; 1, Occasionally; 2, Quite often; 3, Very often: 0, I take just as much care as ever; 1, I may not take quite as much care; 2, I don't take as much care as I should; 3, Definitely: 0, Not at all; 1, Not very much; 2, Quite a lot; 3, Very much indeed: 0, As much as I ever did; 1, Rather less than I used to; 2, Definitely less than I used to; 3, Hardly at all: 0, Not at all; 1, Not very often; 2, Quite often; 3, Very often indeed; 0, Often; 1, Sometimes; Not often; 3, Very seldom). The reversal items are recalculated. Thus, a range of 0–7 points on each section is considered as 'none', 8–10 points as 'doubtful', and 14–21 points as 'definite' [28]. The reliability and validity of the Japanese version of the HADS have been established [29]; in that study, the Cronbach alpha coefficient for the HADS anxiety subscale was 0.8, and that for the depression subscale was >0.5. The present study's questionnaire was cross-sectionally measured as of 11–12 August 2020, and the participants were asked to answer the questions retrospectively, reflecting on three time points: before the issuance of the lockdown order on 7 April 2020, during the lockdown (7 April–25 May 2020), and after the lockdown (25 May–11 or 12 August 2020). The Cronbach alpha values for the HADS-Depression scale in the present study were 0.75, 0.73, and 0.75 for the 'before', 'during', and 'after' the lockdown order periods, respectively. Likewise, the Cronbach alpha values for the HADS-Anxiety scale in the present study were 0.83, 0.82, and 0.83 for the same respective periods.

2.3.2. Subjective Stress Question

We asked our participants about the degree of subjective stress they experienced by using the following question: "How stressed did you feel at the following times: before, during, and after the COVID lockdown was implemented." The participants' responses were rated on a 5-point scale: "1, Not at all; 2, Generally not stressed; 3, Undecided; 4, A little; and 5, A lot." The Cronbach's alpha value of "Subjective stress" in the present study was 0.89.

2.3.3. Stigma Questions for Hikikomori (SQH)

We developed a new scale to assess stigma toward hikikomori, which we named "Stigma Questions for Hikikomori (SQH)". This scale was created in reference to the Perceived Devaluation and Discrimination Scale devised by Link [24–26], which is one of the most frequently used scales for assessing perceptions of social stigma among patients with severe mental disorders. Our scale included the following items: "1. I like staying home for a long time", "2. I like to spend time outside", "3. Most people find it embarrassing to stay at home", "4. Most people think that staying at home is socially acceptable", "5. Most people believe that staying at home is a failure as a person", "6. Most people won't take the opinions of someone who stays at home seriously", "7. Most people would make fun of someone who stays at home", "8. Most people would trust someone who stays at home", and "9. Most people will discriminate against people who stay at home." The participants were asked to answer these questions on a four-point scale (1, I strongly agree; 2, I agree; 3, I partially disagree; 4, I disagree). The reversal items were recalculated.

*2.4. Statistical Analyses*

2.4.1. Study 1: Factor Structure of "Stigma Questions for Hikikomori (SQH)"

To establish the factor structure of the SQH scale, we employed a two-step approach. In the first step, we conducted an exploratory factor analysis (EFA) on the overall sample of 600 participants before the issuance of the lockdown order (T1). The EFA was performed using Promax rotation, and the estimation method and extraction method used were principal axis factoring and Keiser–Guttman rule, respectively. This step allowed us to identify the number of factors present in the SQH scale.

For internal consistency assessment, we calculated Cronbach's alpha coefficients to measure the reliability of the factors. Additionally, we thoroughly examined the maximum and minimum factor loadings of the individual items to assess their contributions to their respective factors. To ensure the robustness of our findings, we calculated the cumulative percentage of the sum of squares of the factor loadings, providing a comprehensive understanding of the variance explained by the identified factors.

In the second step, we conducted a Confirmatory Factor Analysis (CFA) to validate and confirm the factor structure in a different overall sample of 600 participants during the lockdown order (T2). During the CFA, we examined various models, including the one-factor model, two-factor model, second-order factor model, and bi-factor model. We compared model fit statistics to assess their appropriateness and suitability for our study. Although there are no universally agreed-upon cut-off points for fit indices, we employed the following guidelines to assess model fit. Adequate or good fit was indicated by a Comparative Fit Index (CFI) or Tucker–Lewis Index (TLI) greater than or equal to 0.90, as well as a root mean square error approximation (RMSEA) less than or equal to 0.08 [30,31]. For model-to-model comparison, we relied on the Akaike information criterion (AIC) [31]. An insignificant chi-square test of model fit result indicated good fit.

The data were meticulously analyzed using SPSS version 27 and SPSS AMOS version 28 (IBM SPSS Statistics, IBM, Armonk, NY, USA) to ensure accuracy and validity in our statistical results.

2.4.2. Study 2: Repeated Measured Two-Way ANOVA for the Stigma Score, Subjective Stress Score, HADS-Depression Score, and HADS-Anxiety Score

We performed a repeated measured two-way analysis of variance (ANOVA) to examine the interaction effects between time (before/during/after) and group (definite/possible) for each scale: SQH, subjective stress, HADS-Depression, and HADS-Anxiety. In groups in which interaction effects were significant, simple main effects were then further examined and post-hoc tests were conducted. Otherwise, in groups in which interaction effects were not significant, each factor's main effect was examined as-is, and Bonferroni-corrected post-hoc tests were conducted. The data were analyzed using the software described above.

### 2.4.3. Study 3: Descriptive Analysis of Coping Methods during the Lockdown Period

We extracted nouns from the participants' answers to a free-text question ("Please let us know how you coped with difficulties during the lockdown period") for a morphological analysis. Multiple answers were allowed to this question. We ranked the nouns according to their frequency of occurrence. We visualized the results using a "word cloud" with larger font sizes for more highly ranked items. We used Python 6.3 and the Japanese morphological analysis engine MeCab [32] for this analysis.

## 3. Results

### 3.1. Participants' Characteristics

The participants' sociodemographic characteristics are summarized in Table 1. Of the 600 participants, 403 were male and 197 were female; 407 were in their 20s to 40s, accounting for roughly two-thirds of all participants; 193 were in their 50s, accounting for about one-third. There was no significant difference in the proportion of participants who experienced hikikomori by sex ($p = 0.54$, df = 1, $\varphi$ [phi] = 0.54). However, there was a significant difference in the proportion of participants who experienced hikikomori according to age ($p = 0.02$, df = 3, Cramer V = 0.02).

**Table 1.** Characteristics of respondents.

| Variable | Definite Hikikomori | | Possible Hikikomori | | |
|---|---|---|---|---|---|
| | N | % | n | % | |
| gender | | | | | |
| male | 204 | 66.0% | 199 | 68.4% | df = 1 |
| female | 105 | 34.0% | 92 | 31.6% | Pearson χ2 = 0.54, phi = 0.54 |
| age group | | | | | |
| 20–29 | 9 | 3.0% | 8 | 2.7% | |
| 30–39 | 69 | 22.3% | 60 | 20.6% | df = 3 |
| 40–49 | 149 | 48.2% | 112 | 38.5% | Pearson χ2 = 0.02, Cramer V = 0.02 |
| 50–59 | 82 | 26.5% | 111 | 38.2% | |

### 3.2. Study 1: Factor Structure of Stigma Questions for Hikikomori (SQH)

We conducted a factor analysis of the SQH using the data regarding the period before the issuance of the lockdown order (T1). The EFA results revealed the presence of two factors, which were verified to have a strong relationship with the data (Table 2). The Kaiser–Meyer–Olkin (KMO) result was 0.84, and Barlett's test yielded a significant result ($p < 0.001$), indicating a significant difference from the identity matrix. These findings support the validity of administering the structural analysis.

**Table 2.** Results of the factor analyses (main factor method, Promax rotation) of Stigma Questions for Hikikomori (SQH) scale.

| Hikikomori negative α = 0.92 | Mean | SD | F1 | F2 | Communality |
|---|---|---|---|---|---|
| 3. Most people won't take the opinions of someone who stays at home seriously | 2.28 | 0.96 | 0.92 | 0.03 | 0.86 |
| 5. Most people believe that staying at home is a failure as a person | 2.24 | 0.89 | 0.89 | 0.01 | 0.79 |
| 6. Most people will discriminate against people who stay at home | 2.24 | 0.92 | 0.83 | 0.03 | 0.70 |
| 7. Most people would make fun of someone who stays at home | 2.40 | 0.92 | 0.81 | −0.04 | 0.64 |
| 9. Most people find it embarrassing to stay at home | 2.13 | 0.87 | 0.74 | −0.04 | 0.54 |
| Hikikomori affirmative α = 0.71 | | | | | |
| 4. Most people would trust someone who stays at home | 2.91 | 0.73 | −0.020.00 | 0.740.75 | 0.54 |
| 8. Most people think that staying at home is socially acceptable | 2.87 | 0.73 | 0.00 | 0.75 | 0.57 |

For all of the SQH items presented, the Shapiro-Wilk test's *p*-value was <0.01.

The Cronbach's alpha coefficients for the first and second factors were 0.92 and 0.72, respectively. We named the first factor the "hikikomori negative" scale (including questions 3, 5, 6, 7, and 9) and the second factor the "hikikomori affirmative" scale (including questions 4 and 8). The maximum and minimum factor loadings of the items that were included

in the first factor were 0.92 and 0.74, respectively, and the maximum and minimum factor loading values for the items included in the second factor were 0.75 and 0.74, respectively. The cumulative % of the sum of squares of the loadings of the first factor was 51.2%, and that for the second factor was 66.2%. These statistical results are shown in Table 2.

As depicted in Figure 1 and Table 3, the Confirmatory Factor Analysis (CFA) conducted during the issuance of the lockdown order (T2) revealed that the second-order factor model for SQH provided the most suitable fit compared to other models. Although the two-factor model yielded similar statistical results, we opted for the second-order factor model (Figure 1, Table 3): $\chi 2$ = 106.473, df = 13, $p < 0.01$, Comparative Fit Index (CFI) = 0.960, Normed Fit Index (NFI) = 0.955, Tucker–Lewis Index (TLI) = 0.936, Root-Mean-Square Error of Approximation (RMSEA) = 0.110, Incremental Fit Index (IFI) = 0.960, Akaike's Information Criterion (AIC) = 136.473, and Expected Cross-Validation Index (ECVI) = 0.228.

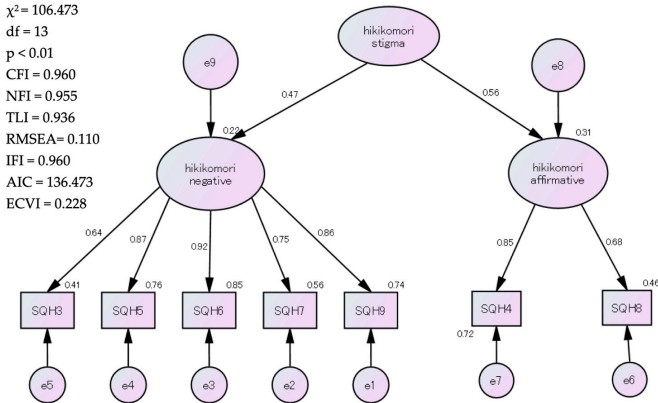

**Figure 1.** Confirmatory factor analysis for stigma Questions for Hikikomori (SQH).

**Table 3.** Model fit statistics.

| Model | CFI | NFI | TLI | RMSEA | IFI | AIC | ECVI |
|---|---|---|---|---|---|---|---|
| One factor | 0.866 | 0.861 | 0.799 | 0.194 | 0.867 | 357.482 | 0.597 |
| Two factor | 0.960 | 0.955 | 0.936 | 0.110 | 0.960 | 136.473 | 0.228 |
| Second order factor | 0.960 | 0.955 | 0.936 | 0.110 | 0.960 | 136.473 | 0.228 |

CFI: comparative fit index, NFI: normed fit index, RMSEA: root-mean-square error of approximation, IFI: incremental fit index, AIC: Akaike's information criterion, ECVI: expected cross-validation index.

Despite the relatively high RMSEA value, we thoroughly considered other fit indices, and overall, the second-order factor model was deemed the most appropriate for our data. Model fit statistics for the one-factor model, two-factor model, second-order factor model, and bi-factor model were compared, and the results are presented in Table 3 (Note: In the bi-factor model, the maximum number of iterations has been reached, and the solution did not converge).

*3.3. Study 2: Two-Way Mixed ANOVA for SQH Hikikomori Negative Scores, SQH Hikikomori Affirmative Scores, Subjective Stress Scores, HADS-Depression Scores, and HADS-Anxiety Scores*

We first confirmed the normal distribution of each item on the questionnaire by examining the corresponding histograms. However, due to the small sample size, we were unable to verify the normal distribution through the Shapiro–Wilk test. It is thus crucial to interpret the following results with caution, considering this limitation.

Table 4 presents the results of the analysis performed to determine the correlations between the differences in the items of the SQH at the three periods (before, during, and after the lockdown) and the differences in psychiatric symptoms (HADS-Depression, HADS-Anxiety, Subjective Stress) at these periods. The analysis revealed either no correlation or only a mild negative correlation between stigma and psychiatric symptoms (Table 4).

**Table 4.** The correlations between the differences in the items of the SQH by period and the differences in psychiatric symptoms (HADS-Depression, HADS-Anxiety, Subjective Stress) by period.

| Variables | M | SD | 1 | 2 | 3 | 4 | 5 | 6 | 7 | 8 | 9 | 10 | 11 | 12 | 13 | 14 | 15 | 16 | 17 | 18 |
|---|---|---|---|---|---|---|---|---|---|---|---|---|---|---|---|---|---|---|---|---|
| 1 SQH (total score) (during-before) | −2.70 | 4.19 | 1.00 | | | | | | | | | | | | | | | | | |
| 2 SQH (total score) (after-before) | −2.39 | 3.74 | 0.932 ** | 1.00 | | | | | | | | | | | | | | | | |
| 3 SQH (total score) (after-during) | 0.31 | 1.52 | −0.463 ** | −0.112 ** | 1.00 | | | | | | | | | | | | | | | |
| 4 SQH (hikikomori negative) (during-before) | −1.75 | 3.06 | 0.967 ** | 0.905 ** | −0.440 ** | 1.00 | | | | | | | | | | | | | | |
| 5 SQH (hikikomori negative) (after-before) | −1.57 | 2.77 | 0.898 ** | 0.963 ** | −0.106 ** | 0.935 ** | 1.00 | | | | | | | | | | | | | |
| 6 SQH (hikikomori negative) (after-during) | 0.18 | 1.09 | −0.432 ** | −0.091 * | 0.966 ** | −0.430 ** | −0.081 * | 1.00 | | | | | | | | | | | | |
| 7 SQH (hikikomori affirmative) (during-before) | −0.94 | 1.45 | 0.844 ** | 0.780 ** | −0.407 ** | 0.679 ** | 0.617 ** | −0.339 ** | 1.00 | | | | | | | | | | | |
| 8 SQH (hikikomori affirmative) (after-before) | −0.82 | 1.29 | 0.765 ** | 0.819 ** | −0.095 * | 0.606 ** | 0.635 ** | −0.088 * | 0.927 ** | 1.00 | | | | | | | | | | |
| 9 SQH (hikikomori affirmative) (after-during) | 0.12 | 0.54 | −0.426 ** | −0.129 ** | 0.856 ** | −0.366 ** | −0.132 ** | 0.692 ** | −0.457 ** | −0.090 * | 1.00 | | | | | | | | | |
| 10 Subjective stress (during-before) | 0.51 | 1.06 | −0.107 ** | −0.078 | 0.103 * | −0.057 | −0.026 | 0.094 * | −0.186 ** | −0.167 ** | 0.099 * | 1.00 | | | | | | | | |
| 11 Subjective stress (after-before) | 0.51 | 1.03 | −0.079 | −0.069 | 0.05 | −0.044 | −0.029 | 0.049 | −0.137 ** | −0.136 ** | 0.042 | 0.845 ** | 1.00 | | | | | | | |
| 12 Subjective stress (after-during) | 0 | 0.58 | 0.052 | 0.019 | −0.097 * | 0.027 | −0.003 | −0.083 * | 0.094 * | 0.061 | −0.104 * | −0.316 ** | 0.241 ** | 1.00 | | | | | | |
| 13 HADS-depression (during-before) | 0.90 | 2.11 | <span style="color:red">0.215 **</span> | −0.192 ** | 0.118 ** | −0.182 ** | −0.165 ** | 0.090 * | <span style="color:red">−0.236 **</span> | −0.201 ** | 0.149 ** | 0.244 ** | 0.210 ** | −0.07 | 1.00 | | | | | |
| 14 HADS-depression (after-before) | 0.77 | 2.04 | −0.136 ** | −0.174 ** | −0.051 | −0.103 * | −0.140 ** | −0.067 | −0.175 ** | −0.200 ** | −0.008 | 0.150 ** | 0.207 ** | 0.096 * | 0.837 ** | 1.00 | | | | |
| 15 HADS-depression (after-during) | 0.12 | 1.18 | 0.147 ** | 0.043 | <span style="color:red">−0.298 **</span> | 0.146 ** | 0.052 | <span style="color:red">−0.276 **</span> | 0.117 ** | 0.013 | <span style="color:red">−0.278 **</span> | −0.176 | −0.017 | 0.290 ** | −0.336 ** | 0.234 ** | 1.00 | | | |
| 16 HADS-anxiety (during-before) | 1.00 | 2.60 | −0.126 ** | −0.115 ** | 0.065 | −0.075 | −0.052 | 0.077 | <span style="color:red">−0.206 **</span> | <span style="color:red">−0.220 **</span> | 0.027 | 0.464 ** | 0.402 ** | −0.129 ** | 0.498 ** | 0.412 ** | −0.176 ** | 1.00 | | |
| 17 HADS-anxiety (after-before) | 0.88 | 2.40 | −0.025 | −0.042 | −0.036 | 0.011 | 0.006 | −0.014 | −0.093 * | −0.135 ** | −0.073 | 0.363 ** | 0.425 ** | 0.096 * | 0.411 ** | 0.505 ** | 0.140 ** | 0.833 ** | 1.00 | |
| 18 HADS-anxiety (after-during) | 0.11 | 1.46 | 0.185 ** | 0.136 ** | −0.175 ** | 0.151 ** | 0.104 * | −0.159 ** | 0.214 ** | 0.169 ** | −0.168 ** | −0.229 ** | −0.016 | 0.388 ** | −0.211 ** | 0.099 * | 0.545 ** | −0.409 ** | 0.163 ** | 1.00 |

** $p < 0.001$, * $p < 0.01$, SQH: Stigma Questions for Hikikomori, HADS: Hospital Anxiety and Depression Scale, Red colored number means that there are mild negative correlations between stigma and psychiatric symptoms.

The mean levels of SQH total score by the hikikomori classification were as follows: definite hikikomori (before: Mean [**M**] = 17.91, standard deviation [**SD**] = 4.56, during: **M = 14.93, SD = 4.12**, after: **M = 15.20, SD = 4.04**), possible hikikomori (before: **M = 16.18, SD = 4.01**, during: **M = 13.77, SD = 3.57**, after: **M = 14.12, SD = 3.42**). The main effect was evaluated as-is because there was no significant interaction effect between the two factors. The main effect of SQH total score at each time period was significant (**F = 19.13**, *p* < **0.001**, $\eta^2$ = **0.03**; Table 5). The Bonferroni post-hoc analysis revealed significant differences in scores between each pair of time periods (Figure 2). The hikikomori classification was also a significant factor (**F = 22.15**, *p* < **0.01**, $\eta^2$ = **0.04**; Table 5).

**Table 5.** Means and statistics of the psychological scales by definite/possible hikikomori group and COVID lockdown time period.

| | Before | | During | | After | | Time | | Group | | Time×Group | |
|---|---|---|---|---|---|---|---|---|---|---|---|---|
| | **M** | **SD** | **M** | **SD** | **M** | **SD** | **F** | $\eta^2$ | **F** | $\eta^2$ | **F** | $\eta^2$ |
| SQH Total score | | | | | | | | | | | | |
| Definite | 17.91 | 4.56 | 14.93 | 4.12 | 15.20 | 4.04 | 19.13 ** | 0.03 | 22.15 ** | 0.04 | 2.84 | 0.05 |
| Possible | 16.18 | 4.01 | 13.77 | 3.57 | 14.12 | 3.42 | | | | | | |
| SQH Hikikomori negative | | | | | | | | | | | | |
| Definite | 12.04 | 3.99 | 10.05 | 3.45 | 10.22 | 3.44 | 182.25 ** | 0.23 | 21.95 ** | 0.04 | 3.91 * | 0.01 |
| Possible | 10.05 | 3.45 | 8.98 | 3.11 | 9.17 | 3.08 | | | | | | |
| SQH Hikikomori affirmative | | | | | | | | | | | | |
| Definite | 5.87 | 1.26 | 4.88 | 1.46 | 4.98 | 1.39 | 231.22 ** | 0.28 | 0.91 | 0.002 | 1.12 | 0.002 |
| Possible | 5.69 | 1.32 | 4.79 | 1.55 | 4.95 | 1.46 | | | | | | |
| Subjective Stress | | | | | | | | | | | | |
| Definite | 11.05 | 4.39 | 11.88 | 4.19 | 11.84 | 4.36 | 144.13 ** | 0.19 | 348.37 ** | 0.102 | 68.01 ** | 0.39 |
| Possible | 8.87 | 4.45 | 9.84 | 4.35 | 9.63 | 4.43 | | | | | | |
| HADS-Depression | | | | | | | | | | | | |
| Definite | 11.05 | 4.39 | 11.88 | 4.19 | 11.84 | 4.36 | 85.42 ** | 0.13 | 38.52 ** | 0.06 | 0.72 | 0.01 |
| Possible | 8.87 | 4.45 | 9.84 | 4.35 | 9.63 | 4.43 | | | | | | |
| HADS-Anxiety | | | | | | | | | | | | |
| Definite | 8.64 | 4.69 | 9.44 | 4.76 | 9.38 | 4.86 | 74.47 ** | 0.11 | 87.09 ** | 0.13 | 2.89 | 0.01 |
| Possible | 5.13 | 3.91 | 6.34 | 4.27 | 6.18 | 4.21 | | | | | | |

** *p* < 0.001, * *p* < 0.05. SQH: Stigma Questions for Hikikomori, HADS: Hospital Anxiety and Depression Scale.

The mean levels of hikikomori negative scores by the hikikomori classification were as follows: definite hikikomori (before: Mean [**M**] = 12.04, standard deviation [**SD**] = 4.00, during: **M = 10.05, SD = 3.45**, after: **M = 10.22, SD = 3.44**), possible hikikomori (before: **M = 10.49, SD = 3.82**, during: **M = 8.98, SD = 3.11**, after: **M = 9.17, SD = 3.08**). There was a significant interaction effect between the hikikomori negative scores in each time period and each hikikomori classification. Further examination revealed a simple main effect in stigma scores between each time period (**F = 182.25**, *p* < **0.001**, $\eta^2$ = **0.23**; Table 5, Figure 2) and hikikomori class (**F = 21.95**, *p* < **0.001**, $\eta^2$ = **0.04**; Table 5, Figure 2).

The mean levels of hikikomori affirmative scores by hikikomori classification were as follows: definite hikikomori (before: **M = 5.87, SD = 1.26**, during: **M = 4.88, SD = 1.46**, after: **M = 4.98, SD = 1.40**) and possible hikikomori (before: **M = 5.69, SD = 1.32**, during: **M = 4.79, SD = 1.55**, after: **M = 4.95, SD = 1.46**). The main effect was evaluated as-is because there was no significant interaction effect between the two factors. The main effect of hikikomori affirmative scores at each time period was significant (**F = 231.22**, *p* < **0.001**, $\eta^2$ = **0.28**; Table 5). The Bonferroni post-hoc analysis revealed significant differences in scores between each pair of time periods (Figure 2), but there was no significant main effect of the hikikomori classification (**F = 0.91**, *p* = **0.34**, $\eta^2$ = **0.002**; Table 5).

The mean level of subjective stress scores by hikikomori classification were as follows: definite hikikomori (before: **M = 3.80, SD = 0.78**, during: **M = 3.96, SD = 0.83**, after: **M = 3.98, SD = 0.81**), possible hikikomori (before: **M = 1.95, SD = 0.98**, during **M = 2.84, SD = 1.36**, after **M = 2.82, SD = 1.32**). The main effect was evaluated as-is, because there was no significant interaction effect. The main effect of the subjective stress score in each

time period was significant (**F = 144.13**, *p* **< 0.001**, η² **= 0.19;** Table 5). The Bonferroni post-hoc analysis revealed significant differences in scores between each pair of time periods (Figure 2). The hikikomori classification was also a significant factor (**F = 348.37**, *p* **< 0.001**, η² **= 0.10**; Table 5).

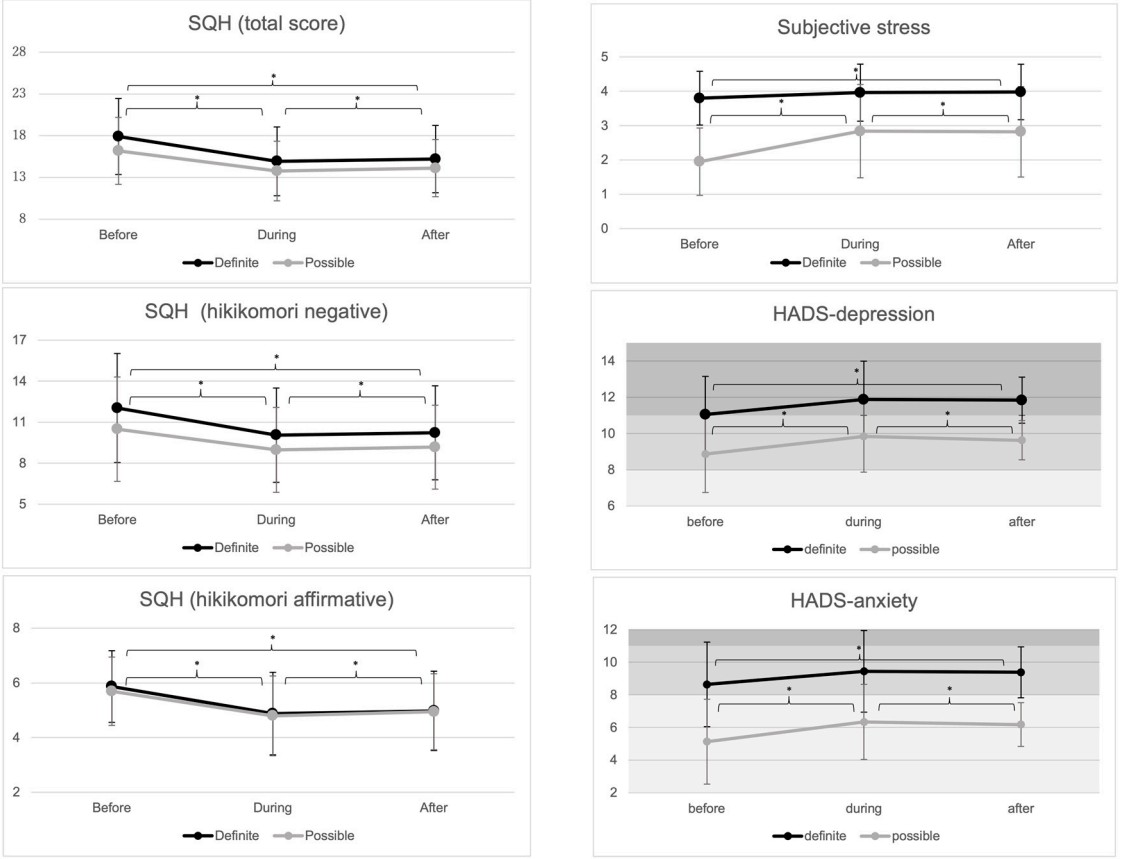

**Figure 2.** Change in scores on psychological scales. Vertical axes: scores on each scale. Horizontal axes: The three time points "before", "during", and "after" the lock down order was implemented. * means *p* < 0.01.

The mean levels of HADS-Depression scores by hikikomori class were as follows: definite hikikomori (before: **M = 11.05**, **SD = 4.39**, during: **M = 11.88**, **SD = 4.19**, after: **M = 11.84**, **SD = 4.36**) and possible hikikomori (before: **M = 8.87**, **SD = 4.45**, during: **M = 9.84**, **SD = 4.35**, after: **M = 9.63**, **SD = 4.43**). The main effect could be evaluated as-is because there was no significant interaction effect. The main effect of the HADS-Depression score at each time period was significant (**F = 85.42**, *p* **< 0.001**, η² **= 0.13**; Table 5). The post-hoc analysis revealed significant differences in scores between each pair of time periods (Figure 2). The hikikomori classification was also a significant factor (**F = 38.52**, *p* **< 0.001**, η² **= 0.06**; Table 5).

The mean level of HADS-Anxiety scores by hikikomori classification were as follows: definite hikikomori (before: **M = 8.64**, **SD = 4.70,** during: **M = 9.44**, **SD = 4.76**, after: **M = 9.38**, **SD = 4.86**) and possible hikikomori (before: **M = 5.13**, **SD = 3.91**, during: **M = 6.34**, **SD = 4.27**, after: **M = 6.18**, **SD = 4.21**). The main effect was evaluated as-is because there was no significant interaction effect between the two factors. The main effect of the HADS-Anxiety score at each time period was significant (**F = 74.47**, *p* **< 0.001**, η² **= 0.11**; Table 5). The Bonferroni post-hoc analysis revealed significant differences in scores between each pair of time periods (Figure 2). There was a significant main effect of hikikomori classification (**F = 87.09**, *p* **< 0.001**, η² **= 0.13**; Table 5).

Because there was a significant difference according to age in the proportion of participants who experienced hikikomori, we also excluded age as a confounding factor in the analysis, but the same results were obtained. There was no significant difference by sex in the proportion of participants who experienced hikikomori, and, because it was not considered a confounding factor, the results were interpreted as they were.

*3.4. Study 3: Descriptive Analysis of Coping Methods during the Lockdown Period*

A total of 237 participants answered the question "Please let us know how you coped with difficulties during the lockdown period", providing a total of 570 responses regarding coping methods (multiple answers were allowed). Terms related to indoor activities that could be engaged in at home were frequently reported, including games, television, hobbies, and cooking. These results suggested that the participants' coping strategies were focused on how to enjoy staying at home and may thus indicate that participants did not have difficulty staying at home per se, but that aspects of the pandemic caused psychological stress. Figure 3 depicts the analysis output in the form of a word cloud (the Japanese terms have been translated into English). The color and location of each word are random, but the font size corresponds to the frequency of each term.

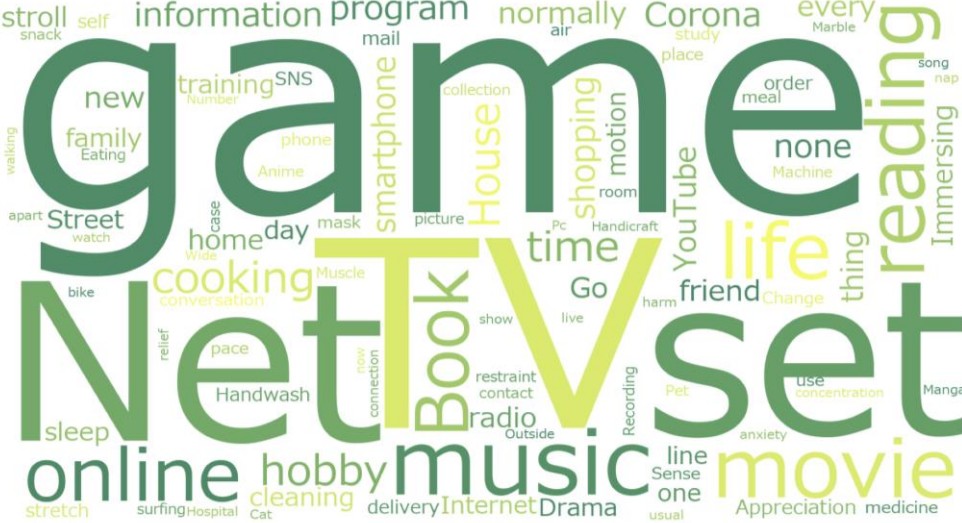

**Figure 3.** The results of the descriptive analysis of the participants' coping methods during the COVID lockdown. The size of each word indicates the frequency of the coping strategy used for definite or possible hikikomori.

## 4. Discussion

Although several studies have reported on the relationship between post-pandemic quarantine and psychological effects [1,33], our study is the first investigation of individuals with definite or possible hikikomori during the COVID-19 outbreak. Because it is difficult to understand the diverse realities of hikikomori under normal conditions before the pandemic occurred, effective measures against the condition had not been devised. The purposes of this study were to (*i*) gain a better understanding of hikikomori, and (*ii*) identify ways to provide effective support to people suffering from this condition.

Our findings revealed that depression and anxiety among people with hikikomori worsened during the COVID-19 pandemic, even though the sense of stigma regarding hikikomori improved. We hypothesized that stigma toward hikikomori would be ameliorated during the COVID-19 outbreak, and that the pandemic would improve the mental state of hikikomori sufferers by creating circumstances in which social distancing was required for all citizens. This hypothesis was not supported by the present study's results, but the results also revealed several unexpected findings that may be useful for understanding and supporting individuals with hikikomori in the future.

We detected a difference in the severity of psychological symptoms between people classified as having definite hikikomori and those classified as having possible hikikomori, with the former exhibiting more severe depression and anxiety. This finding suggests that it is important to accurately assess individuals' hikikomori status and implement targeted interventions for those who fulfill the diagnostic criteria for the condition [26]. We also observed that individuals with definite hikikomori scored above the cut-off score for depression, suggesting the need for more immediate interventions. In order to enhance interventions aimed at addressing depression among individuals who experience hikikomori and social isolation, it may be beneficial to consider the implementation of cognitive-behavioral therapy (CBT), which is widely recognized as an evidence-based treatment approach [34]. Particularly during the pandemic, Internet-based CBT can be valuable as it mitigates the risk of infection transmission [35]. It is worth noting that approaching individuals with hikikomori can be challenging due to their avoidant and narcissistic traits; it might be effective to initially engage with their families and indirectly influence the hikikomori sufferer's behaviors. A study conducted in Japan obtained promising results in terms of improving behavioral issues among individuals with hikikomori through a family-based intervention program [36].

Our results also revealed an increase in subjective stress among the participants with tendencies toward hikikomori after the COVID lockdown was implemented. This stress might have been a factor in the exacerbation of depression and anxiety we observed among the participants with hikikomori. This subjective stress may have been related to frustration with the enforced restrictions on behavior, and/or the fear of COVID-19. Our participants with hikikomori-related traits (definite and possible hikikomori) might have felt these stresses more strongly than people in the general population, potentially exacerbating their depression and anxiety. Thus, although stigma against hikikomori improved, potentially affecting the participants' mood in a positive way, the fear of infection might have caused their mood to worsen.

Several studies reported that fear of COVID-19 can induce mental disturbances [37–39]. Worsening depression was observed among individuals with mental illness as the fear of COVID-19 spread among them [40]. Other studies reported that the capacity for tolerance toward an uncertain future affects mental disorders such as depression, obsessive compulsive disorder, and eating disorder [41,42]. Intolerance regarding uncertainty about the future has been observed to affect mental well-being in this infectious situation and to contribute to the severity of mental disorders such as depression [43,44]. We speculate that our participants with hikikomori-related traits (definite and possible hikikomori) were already highly anxious about their uncertain future. The COVID-19 pandemic might thus be expected to exacerbate these worries and cause additional anxiety. A cross-sectional study conducted in Greece reported that intolerance of uncertainty was a strong predictor of depression during the COVID-19 pandemic, and that fear of infection acted as a mediator for the effect on depression [45].

The results of the present descriptive analysis indicated that the participants with definite or possible hikikomori adopted relatively effective coping strategies during the COVID-19 pandemic. This suggests the value of learning from individuals with hikikomori by examining their behavior as a model. Individuals with hikikomori are often labeled negatively, and considering their behaviors as a positive model might contribute to reducing the stigma against them. We propose that reconsidering the behavior of people with hikikomori in this way may be valuable when approaches for managing issues related to this condition are sought.

However, several limitations of this study should be considered when interpreting the results. Hikikomori is a multifaceted phenomenon, and its understanding and definition can certainly vary based on cultural and academic perspectives. The criteria used in this study may not fully capture the complexity and diversity of hikikomori. In order to enhance the quality of hikikomori diagnoses in the future, it is crucial to adopt a multidirectional approach in clinical research, encompassing brain-imaging analyses, biomedical data, and

other relevant factors. We also used a retrospective survey method in which participants scored the scale items based on their recollection of the situation at the time of the lockdown period, and it is thus not possible to draw causal conclusions from the questionnaire results regarding the factors that exacerbated depression and anxiety among the participants.

It is important to note that the data for our study were collected via a web survey company. Due to this method of data collection, the exclusion/inclusion criteria were not clearly defined, leading to an unbalanced sample in terms of gender representation. This potential imbalance should be taken into consideration when interpreting and generalizing our findings. In addition, in both Study 1 and Study 2, although we visually confirmed the normal distribution of the sample through the examination of the corresponding histograms, the Shapiro-Wilk test did not support the normal distribution assumption. The use of Levene's test also did not provide evidence of the equality of variances among the variables. Moreover, given the prevalence of widespread loneliness during the pandemic, it becomes challenging to differentiate individuals experiencing genuine hikikomori from those who have simply been compelled to stay at home due to their circumstances. Further studies are necessary to deepen our understanding of the psychological changes occurring among people with hikikomori during the pandemic period, which resulted in exacerbations of depression and anxiety.

## 5. Conclusions

During Japan's lockdown due to the COVID-19 pandemic, the stigma against hikikomori improved. The individuals who reported hikikomori traits experienced a worsening of depression and anxiety during the lockdown. The improvement of stigma and the participants' indoor adaptive coping strategies alone could not ameliorate the mental health of those with hikikomori. The individuals with definite hikikomori (i.e., they met all of the diagnostic criteria) exhibited poorer mental health and needed more therapeutic intervention compared to the participants who did not have hikikomori.

**Author Contributions:** Conceptualization: T.O., Y.S., H.T., H.M., M.A., D.S., N.K. and T.A. Methodology: T.O. and H.T. Software: T.O. Validation: Y.S. and H.T. Formal analysis: T.O., Y.S. and H.T. Investigation: T.O., Rakuten Insight, Inc. Resources: T.O., Rakuten Insight. Data curation: T.O., Rakuten Insight. Writing—original draft preparation: T.O. Writing—review and editing: Y.S., H.T. Visualization: T.O.; Supervision: Y.S., H.T. Project administration: H.T. Funding acquisition: H.T., Y.S., M.A., D.S. and N.K. All authors have read and agreed to the published version of the manuscript.

**Funding:** This work was supported by a grant (no. JPMJRX21K2) from the JST RISTEX "SOLVE for SDGs: Preventing Social Isolation & Loneliness and Creating Diversified Social Networks" program, Japan.

**Institutional Review Board Statement:** The study was conducted according to the guidelines of the Declaration of Helsinki, and approved by the Ethics Committee of the University of Tsukuba (Registration No.1546-1).

**Informed Consent Statement:** This research was conducted in accord with the Declaration of Helsinki. The authors assert that all procedures contributing to this work comply with the ethical standards of the relevant national and institutional committees on human experimentation and with the Helsinki Declaration of 1975, as revised in 2008. We respected the participants' right to privacy and obtained informed consent from all participants. This study was approved by the Medical Ethics Committee of the University of Tsukuba (Registration no. 1546-1).

**Conflicts of Interest:** The authors declare no conflict of interest.

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
