# Peer review of "A Survey of Changes in the Psychological State of Individuals with Social Withdrawal (hikikomori) in the Context of the COVID Pandemic"

_covid, doi:10.3390/covid3080082_

Round 1

Reviewer 1 Report

The manuscript by Ogawa et al. " A Survey of Changes in the Psychological State of Individuals with Hikikomori in the Context of the Coronavirus Disease 2019 Pandemic" aims to investigate and examine psychological changes among Japanese individuals with “hikikomori” during COVID 2019 pandemic via three types of questioners at three time points.  The authors found that depression and anxiety become worsen in individuals with “hikikomori” during and after COVID 2019 pandemic, it was particularly observed in “definite hikikomori”. It is an interesting and important approach to mental health welfare in social withdrawal. However minor concerns are raised in the present form of the text. 

Minor concerns

1) Hikikomori is the Japanese term for a person who is absent from school or work and is confined at home for more than 6 months. Although it has attracted international attention in recent years, it is better to describe hikikomori properly and mentioned it as social withdrawal in the title, and the introduction.

2) As the authors pointed out that it is important to accurately assess hikikomori status and implement targeted interventions for individuals that fulfill the diagnostic criteria for the condition, it is not clear between “definite” and “possible” set by Kamba er al (ref 25). Clinical research targeting hikikomori should incorporate a multidirectional approach, which includes brain imaging analyses and biomedical data to improve the quality of diagnosis of hikikomori.

3) This study includes a wide range of ages and both genders. There is a clear gender difference in hikikomori and the symptom of anxiety and depression. These symptoms are also differentially observed in different ages. Hormonal variation and temporal dynamics are associated with age, body condition, gender, and the social environment. Therefore, some data showed a high F value. Please add a reasonable justification to mix genders and a wide range of ages.

4) I am wondering whether the authors’ hypothesis and conclusion about the stigma toward hikikomori during COVID 2019 pandemic. I am also concerned whether the data based on the questionnaire leads to the stigma against hikikomori is improved. In this closed environment such as COVID 2019 pandemic, hikikomori, mental illness patients and elderly people become lonely. It became more difficult to distinguish them from non-hikikomori and created a huge disparity.

This review would require minor revision in terms of language and information provided. 

Author Response

Dear reviewer 1,

Thank you for your sincere and kind comments and highly suggestive advice for the revision. We are very grateful for the time and energy you expended. We have taken your valuable counsel into careful consideration and made the following changes, which are indicated by underlines below. Additionally, we have sought the assistance of a professional translational service to ensure the accuracy and quality of the revision.

Sincerely,

Yuki Shiratori

Reviewer 2 Report

The manuscript "A Survey of Changes in the Psychological State of Individuals with Hikikomori in the Context of the Coronavirus Disease 2019 Pandemic" is an interesting study about the hikikomori symptoms during the pandemic. Although the topic is interesting, the manuscript contains several serious methodological flaws and problems that must be resolved before the manuscript is reconsidered for publication.

1. The introduction is well written and interesting. However, information about the restriction level during the data collection in this study is required to understand the pandemic context. Please visit the world COVID-19 data collections (e.g., https://ourworldindata.org/policy-responses-covid) and demonstrate the information about the number of infection cases, deaths, stringency index, etc.

2. Materials and methods require improvement:

How were the participants invited to the study? 

How many people were excluded from analysis?

What was the response rate?  

It is unclear whether the sample of 600 participants contains only those who met the criteria for the Japanese Cabinet in 2018 quality of life questionnaire (5<), or all people with other response options are also included (1-4).

Please describe the three questions content in details, and all response options, to define hikikomori (Kamba et al.).

Please add reference to Japanese adaptation of the HASD, information about the response scale and its numerical coding for HADS, and Cronbach's alpha for previous research and current study sample. It is unclear whether the respondent answered each question three times? Please add information about the lockdown duration. What does it mean "before", "during" and "after" lockdown order was implemented? What were the restriction levels, cases, deaths, etc. in each period?

What was the Cronbach alpha for "subjective stress" question performed three times retrospectively?

Please add information about the response scale used for the SQH questionnaire (all options to select and its coding).

Statistical analysis needs extension about effect size and its interpretation for ANOVA (e.g. partial eta-square). It is unclear whether the data for each questionnaire met criteria for parametric tests, including normal distribution assumption and equality of variance. Please explain it. 

3. The result section must be improved:

Table 1 should also include more demographic variables (e.g., education, occupational status) and more columns for those with "possible" and "definite" hikikomori symptoms, with results of the chi-square test of independence to examine whether these groups differ in demographics. Please add more stats, including chi-square test, df, p, and effect size (e.g., phi or Cramer's V, depending on the number of categories).

The structural analysis for SQH should include EFA and CFA. Please divide the total sample randomly in two subsamples, one for EFA, and second for CFA.

Please explain which condition was evaluated, "before", "during", or "after" lockdown? Or maybe all three separately?

Please show the initial statistics, namely the Bartlett’s test of sphericity and Kaiser-Meyer-Olkin (KMO) for the EFA sample. 

Please add one column to show the normality test (e.g., Shapiro-Wilk) for each question in table 2. 

Please add CFA analysis to examine the fit measures for the two-factor model, comparing various models, one-factor, two-factor, hierarchical second order, and bi-factor models. Also, component CFA (with AVE, CR, and HTMT) could be helpful to examine whether the SQH questionnaire is a valid instrument.

Reliability should be shown for each repeated measurement (three times for each scale and subscale) for SQH and HADS as well.

Please add the correlation between SQH and depression, anxiety, and stress measures.

It is unclear what type of ANOVA you use. The repeated measured two-way ANOVA should be implemented Time (before, during, and after lockdown) x Group (possible, definite hikikomori).  Please explain it and use tables for each ANOVA test, showing all M and SD for each variable in each condition and level. See the APA style guideline, to learn which statistics should be included and how arranged. Add df for F-test and effect size. 

All figures require improvement, including resolution (is very poor) and way to show the variables and significant differences. Please see the APA style manual to learn how the figure should be prepared to be clear, informative, and transparent.

4. Discussion should be extended in accordance with the new data.

Also, limitations should be improved, by adding the discussion about bias related to unbalanced gender samples, inclusion/exclusion criteria, the form of data collected (online survey), and many other issues.

Please add a practical implication subsection and describe how to prevent hikikomori, what intervention methods should be implemented, and for what targeted groups.

Author Response

Dear reviewer 2,

Thank you for your sincere and kind comments and highly suggestive advice for the revision. We are very grateful for the time and energy you expended. We have taken your valuable counsel into careful consideration and made the following changes, which are indicated by underlines below. Additionally, we have sought the assistance of a professional translational service to ensure the accuracy and quality of the revision.

Sincerely,

Yuki Shiratori

Round 2

Reviewer 2 Report

Dear Authors, I appreciate the effort put into improving the manuscript. Most of my suggestions were well taken into account. However, there are still some problems that prevent the publication of the manuscript in its current form.

1. Please add the reference to the stringency index when it is cited first time (page 5, line 12).

2. Please add the response option for HADS scale to the manuscript, as previously suggested. If "The responses are rated on a 4-point scale", a reader should know what was interpreted score 1, 2, 3, and 4.

3. If HADS was assessed three times, please add Cronbach's Alpha for both Depression and Anxiety subscales in each measurement (three times, for "before", "during", and "after" lockdown), respectively.

4. Please add Cronbach's alpha the "Subjective stress question" to the manuscript (currently you provided the stats in a cover letter!).

5. It is inappropriate to perform the EFA (N = 600) at the same sample as the EFA (n = 300) and CFA (n = 300) in the same measurement. Moreover, it is unclear what sample (T1 or T2) was included in the data analysis for EFA and CFA, and EFA again. Please describe it for transparency and eventual replication of the study. 

6. The sample size of 300 participants is too small for EFA and CFA (a rule of thumb is to collect 10 cases per item, so 700 participants you require to have a stable structure). However, you can easily fix this problem. If the SQH was performed two times, you can use the first measurement (T1) for EFA, while the second (T2) for CFA, instead of dividing the total sample into half. As such, Table 2 should be left, but CFA should be performed for T2 measurement and included into the manuscript instead of current Fig.1.

7. It is unclear what estimation method was used for EFA (e.g., Maximum Residuals, Maximum Likelihood, Principal Axis Factoring, or Ordinary Least Squares, etc.) and what rotation technique (orthogonal or oblique). What extraction method was used (e.g., Keiser criterion or parallel analysis? If yes, which one?). Please describe it exhaustively for better transparency and further replication.

7. As previously suggested, various models should be examined and model fit stats compared in CFA (using the same sample at T2), including one-factor model (one latent variable for all 7 items), two-factor (as currently it is performed), second-order factor model (with one latent variable = hikikomori stigma on the highest level, two factors on a lower level, and 7 items at the lowest level), and bi-factor model (similar to current model, but with also one factor for all 7 items). All model fit stats should be compared in one table.

8. Please add interpretation and references to explain what criteria are used for fit statistics.

9. Table 3 and 4 are not provided, therefore it is unclear what data the Authors reported. However, it is inappropriate to report Fisher-test without dfs and effect size in the manuscript's text. The exemplar of reporting ANOVA can be: F(2, 567) = 4.56, p < 0.001,  η2p = 0.07. Also you should use the other symbols for mean (M = 5.69) and standard deviation (SD = 1.55). Please add Table 3 and 4 to the next version of the manuscript, and correct all statistics in section 3.3.

10. The SQH is a very promising tool for its future use in the Japanese population and to translate it into many languages. Therefore, it is important to report all the critical stats in the current manuscript. Therefore, I strongly encourage the Authors  to improve the manuscript again.
